# Changes in the Use of Antibiotics for Methicillin-Resistant *Staphylococcus aureus* Bloodstream Infections in Children: A 5-Year Retrospective, Single Center Study

**DOI:** 10.3390/antibiotics12020216

**Published:** 2023-01-20

**Authors:** Maria Sole Valentino, Paola Borgia, Virginia Deut, Ines Lorenzi, Paola Barabino, Elisabetta Ugolotti, Marcello Mariani, Francesca Bagnasco, Elio Castagnola

**Affiliations:** 1Department of Woman, Child and General and Specialistic Surgery, L.Vanvitelli Campania University, 80138 Napoli, Italy; 2Department of Pediatrics, Infectious Disease Unit, IRCCS Istituto Giannina Gaslini, 16147 Genova, Italy; 3Department of Neuroscience, Rehabilitation, Ophthalmology, Genetics, Maternal and Child Health (DINOGMI), University of Genoa, 16126 Genova, Italy; 4Pharmacy, IRCCS Istituto Giannina Gaslini, 16147 Genova, Italy; 5Epidemiology and Biostatistics, Scientific Directorate, IRCCS Istituto Giannina Gaslini, 16147 Genova, Italy

**Keywords:** DDD, MRSA, vancomycin, daptomycin, ceftaroline, children

## Abstract

Monitoring antibiotic use in the pediatric population is a challenge, especially when determining a relationship between specific pathogens, infections, and antibiotic use. We retrospectively analyzed the consumption of anti-methicillin-resistant *Staphylococcus aureus* (MRSA) drugs from 2017 to 2021 at Istituto Giannina Gaslini by means of defined daily dose (DDD) adopted for adults by World Health Organization. We observed a statistically significant increase in the use of daptomycin and ceftaroline, combined with a decrease in the use of vancomycin. In the same period, we observed an increase in the proportion of bloodstream infections due to MRSA with vancomycin minimally inhibitory concentration (MIC mg/L) = 1, that represented the 100% of cases in 2021. This aspect was combined with the observation that in the 59% of cases, where vancomycin plasma concentrations were evaluated, it was not possible to achieve a ratio of the 24-h area under the concentration–time curve and MIC (AUC_0–24_/MIC) of vancomycin ≥ 400 mg/L. This study confirms that DDD can be used in pediatrics to monitor antibiotic consumption in relationship with infections epidemiology. Moreover, it describes the presence of vancomycin MIC creep for MRSA also in pediatrics and the difficulties in obtaining effective vancomycin plasma concentrations in children.

## 1. Introduction

*Staphylococcus aureus* is a Gram-positive, coagulase-positive pathogen that frequently colonizes skin, skin glands, and mucous membranes of healthy individuals [1,2]. Methicillin-resistant *Staphylococcus aureus* (MRSA) presents an altered penicillin-binding protein (PBP) and is associated with decreased affinity for most semisynthetic penicillins [3]. Therefore, vancomycin has been the agent of choice to treat MRSA infections for many years [4]. Susceptibility to vancomycin is defined in presence of minimally inhibitor concentrations (MIC) ≤ 2 mg/L [5], but presence of MIC values > 1 mg/L have been associated with treatment failure, with the consequent recommendation to change therapy in presence of these MIC values [6]. In the past few years, new antibiotics including ceftaroline, daptomycin, dalbavancin, tedizolid, oritavancin, and telavancin, have been approved for treatment for staphylococcal infections in adults, including MRSA [7]. Vancomycin, however, remains the most widespread agent for treatment of infections due to MRSA in children, despite its decreasing susceptibility and toxicity [4,6]. Monitoring of epidemiology of antibiotic resistant bacterial infections and of antibiotic consumption are pivotal for a correct antimicrobial stewardship program especially in an era of increasing antibiotic resistance. This may be of utmost importance in countries such as Italy where the incidence of antibiotic-resistant infections is high, and has worsened during COVID-19 pandemics [8], although this aspect has not been observed in Italian children [9].

Although antibiotic consumption monitoring is one of many aspects of antimicrobial stewardship, until today, there is no consensus on the best approximation method in pediatrics. Recently, in our institution, a defined daily dose (DDD), as defined by the World Health Organization (WHO) Collaborating Center for Drug Statistics Methodology, based system for monitoring antibiotic consumption has been implemented [10]. This procedure should not be considered as an exact picture of actual use, but only a fixed unit of measurement for a not time-consuming estimate of drug consumption. Therefore, it could be used to monitor the use of drugs, to assess patterns in drug utilization, to perform comparisons of drug use between different settings and between different drugs also in relationship with possible changes in infections epidemiology.

The aim of the present study was to describe the consumption of anti MRSA antibiotics in bloodstream infections (BSIs) during a 5-year period in children admitted at a single center, and its relationship with patterns in MRSA antibiotic susceptibility and the possibility of achieving a therapeutic plasma concentration for vancomycin.

## 2. Materials and Methods

The IRCCS Istituto Giannina Gaslini (IGG), Genoa, Italy, is a tertiary care children’s hospital in northern Italy serving as local pediatric hospital for the Genoa area, but representing a tertiary care referral hospital for Italy and many foreign countries.

From January 2017 to December 2021, the consumption of anti MRSA antibiotics was retrospectively evaluated considering the amount of drug (in grams) dispensed by IGG Pharmacy service to the whole hospital wards of vancomycin, teicoplanin, daptomycin, ceftaroline, linezolid (intravenous or oral) [10]. The number of dispensed DDDs of anti MRSA antibiotics was analyzed within an antibiotic stewardship program. For each drug, the number of administered DDDs for year was calculated dividing the amount of dispensed drug by the DDD indicated by the WHO Collaborating Center for Drug Statistics Methodology [11].

The following data were extracted from the IGG Laboratory of Microbiology database and anonymously analyzed: positive blood cultures for MRSA with vancomycin MIC evaluated by means of Sensititre (Thermo Scientific, Thermo Fisher Diagnostic, Landsmeer, The Netherlands), vancomycin plasma concentrations corresponding to the same hospital admission code, if performed within 5 days following a positive blood culture, age at diagnosis of MRSA BSI, ward of admission as surrogate of underlying disease, and concomitant infection by SARS-CoV-2 in the period 2020–2021.

### 2.1. Standard of Care for MRSA Bacteremia

During the study period, vancomycin at a dose of 40 mg/kg/day divided in 2–4 doses or administered in 24 h continuous infusion (with a 10 mg/kg loading dose) was the standard initial therapy in absence of nephropathy or known previous severe adverse events of vancomycin. Plasma concentrations of the drug were routinely evaluated by means of commercial immunoassay, with dose increase up to 60 mg/kg/day if needed in order to obtain a ratio of the 24 h area under the concentration–time curve and MIC (AUC_0–24_/MIC) of vancomycin ≥ 400, calculated as previously described [10], for intermittent or continuous infusion and a through level < 25 mg/L or AUC_0–24_ < 600.

If adequate vancomycin level could not be achieved or in presence of other contraindications, teicoplanin, daptomycin, ceftaroline, and linezolid were administered.

For teicoplanin, the dosage was 10 mg/kg (with a load) (maximum 600 mg) q24h; for daptomycin, 10 mg/kg q24h; for ceftaroline, 12 mg/kg (maximum 600 mg) q8h; and for linezolid, 10 mg/kg q8h in patients < 12 years old or q12h (maximum 600 mg) for older subjects [12].

Combination therapies (e.g., daptomycin + ceftaroline) were performed in cases of severe infections requiring prolonged intravenous therapy in order to improve effectiveness and reduce the risk of resistance selection [13].

### 2.2. Statistical Analysis

Data were reported as absolute number and percentages for categorical variables and as median and 1st and 3rd quartile (interquartile range, IQR) for the continuous one. Percentages of MRSA MIC value, stratified as ≤0.5 mg/L or ≥1 mg/L, were calculated and trend over time (5-year period) was assessed by the chi-square statistic for the trend of proportions [14].

The relationship between dispensed DDDs and year of observation (time) was analyzed by means of nonparametric local-constant Li-Racine kernel regression with a bandwidth of 0.5 [15]. The nonparametric Kendall rank correlation coefficient was applied to measure the ordinal association between dispensed DDDs and years. The nonparametric test for trend across ordered groups developed by Cuzik [16] was applied for trend of DDD values across years.

All tests were two-tailed and a *p* value < 0.05 was considered statistically significant. All analyses were performed using Stata (StataCorp. Stata Statistical Software, Release 16.1 College Station, TX, USA, StataCorporation, 2019).

This study was conducted in accordance with the Helsinki Declaration. According to Italian legislation, the study did not need ethical approval, as it was a purely observational retrospective study on routine collected anonymous data. Moreover, informed consent for participate in the study was not required since retrospective data were obtained by an anonymous microbiology database. In any case, consent to completely anonymous use of clinical data for research/epidemiological purposes is requested by clinical routine at the time of admission/diagnostic procedure.

## 3. Results

During the study period, the median total vancomycin consumption was 1903.5 g, IQR (1603–2363.5) corresponding to a median number of DDDs of 952, IQR (802–1182), Table 1. As shown in Figure 1, vancomycin consumption decreased over time, from 1244 DDDs in 2017 to 794 DDDs in 2021, for a percentage decrease of the 36%. This decreasing trend (Kendall’s rank correlation coefficient equal to −0.8) did not reach a statistical significance (*p* = 0.086), Table 1.

On the contrary, a statistically significant increase in the use of daptomycin and ceftaroline across years (for both antibiotics, Kendall’s rank correlation coefficient equal to 1.0 and *p* = 0.027) was observed, Table 1 and Figure 1. In particular, the median total of daptomycin DDDs was 387, IQR (344–387), changing from 70 DDDs in 2017 to 807 DDDs in 2021 and the median total of ceftaroline DDDs was 77, IQR (41–105), from 0 in 2017 to 312 DDDs in 2021, Table 1.

As shown in Figure 1, no trend over time can be assumed for teicoplanin and linezolid. The median total of teicoplanin DDDs was 633, IQR (429–645) and the median total of linezolid DDDs was 198, IQR (164–223), Table 1.

A total of 45 BSIs due to MRSA were diagnosed in the study period in patients with a median age of 12 months, IQR (1–144). The most frequent wards of admission were the cardiovascular surgery, 11 cases (24.4%), and the intensive care unit, 10 cases (22.2%). Parity of cases, 6 (13.3%), was in hemato/oncology unit and in units admitting other immunocompromission/solid organ transplantation patients. The remaining 12 MRSA BSIs were diagnosed in neonatal or general pediatrics units, 8 (17.8%), and 4 (8.9%) in general surgery.

In 11 (24.4%) cases the MIC for vancomycin was ≤0.5 mg/L, while in the remaining 34 (75.6%) it was equal to 1 mg/L, Table 1. No case of vancomycin MIC ≥ 2 mg/L was observed. The yearly number of MRSA BSIs stratified by MIC values for vancomycin are reported in Table 1. The increase across years in the proportions of MRSA with vancomycin MIC equal to 1 mg/L was statistically significant (*p* = 0.017) and it is noteworthy that in 2021 no isolated strain had MIC ≤ 0.5 mg/L.

Figure 2 shows the time distribution of the number of daptomycin and ceftaroline DDDs and of the percentage of MRSA BSIs, according to vancomycin MIC. A relationship can be assumed between the increased consumption of these two antibiotics over time and the change in the epidemiology of MRSA BSIs.

Vancomycin plasma concentrations were available in 32 (71.1%) BSIs and the median concentration was 14.3 mg/L, IQR (10–18.8). In 13 (40.6%) cases, plasma concentrations allowed an AUC_0–24_/MIC ratio ≥ 400 mg/L, while in 19 (59.4%) vancomycin had to be substituted with another drug because of persistently insufficient AUC_0–24_/MIC ratio.

No case was associated with SARS-CoV-2 infection. Moreover, analysis of hospital causes of death during the study period did not evidence any death due to MRSA bacteremia.

## 4. Discussion

This study allowed to identify different aspects useful for antimicrobial stewardship programs in a tertiary care pediatric hospital.

First of all, we confirmed that DDD indicated by the World Health Organization [10,11] can be adopted also in pediatrics to quantify antibiotic utilization and, in particular, to monitor the use over time. Evaluation of drug consumption is an essential component for a correct antibiotic stewardship program that can both optimize the treatment of infections and reduce adverse events associated with antibiotic use. DDD can be considered a standard to quantify antibiotic use in different settings and contexts providing a fixed measurement unit to perform comparisons. We adopted this method to estimate the consumption of anti-MRSA antibiotics DDD delivered by hospital pharmacy in the period 2017–2021 at IGG. This analysis allowed us to observe a statistically significant increase in the use of daptomycin and ceftaroline, associated with a non-significant decrease in vancomycin consumption. Following this observation, we analyzed the epidemiology of MRSA BSI to understand whether these changes in antibiotic consumption had been driven by modifications in the epidemiology of MRSA, particularly with regard to antibiotic susceptibility. Indeed, we observed a statistically significant increase over time in the proportions of MRSA with higher vancomycin MIC, even if in absence of fully resistant strains.

The gradual increase in the value of vancomycin MIC for *S. aureus* has been reported as MIC creep [4,6], with increasing evidence in the literature that vancomycin maybe ineffective against increasing proportion of isolates with MICs between 1 and 2 mg/L [6]. Strains of MRSA with high vancomycin MICs are associated with poor outcomes especially in patients with bacteremia and deep tissue infections [4,17,18,19,20]. Even if limited to MRSA BSI, this study shows that vancomycin MIC creeping is a diffuse phenomenon that led to changes in antibiotic use also in pediatrics.

We evaluated vancomycin plasma concentrations in more than two-third of cases and also found difficulties in achieving effective concentrations, and that presence of MRSA strains with increased MIC could have played a role in the increased use of daptomycin and ceftarline. The need for higher vancomycin doses to achieve a ratio of AUC_0–24_/MIC ≥ 400 in pediatric patients has been reported, with the recommendation of changes in dosing strategies in order to achieve target pharmacokinetic/pharmacodynamics parameters [20]. Our data confirm this observation and indicates that frequently the target values cannot be reached and therefore alternative drugs are needed for treatment of MRSA BSI.

During the study period, we also observed a non-significant decrease in vancomycin use. This is probably related with the fact that vancomycin is used also for treatment of other Gram-positive, e.g., coagulase-negative staphylococci, beta-lactam-resistant infections, and even empirically in some clinical conditions. So, the use of alternative antibiotics for MRSA BSI probably changed only marginally the overall use of vancomycin. The use of teicoplanin and linezolid was marginal, and therefore we cannot estimate changes during time.

COVID-19 has been an important driver of infections due to antibiotic-resistant bacteria, including MRSA, in Italy [8]. However, it has not been a major problem in pediatrics, as documented by observation performed in our hospital [21] and an Italian multicenter epidemiologic study [9], and by the observation that none of the cases we collected were associated with SARS-CoV-2 infection.

Our study also indicates that currently knowledge of the local epidemiology of antibiotic susceptibility is critical for proper initial management of patients with suspected/confirmed infections such as BSI. Rapid microbiological diagnosis may be important for early identification of colonized patients with subsequent adoption of prophylactic strategies. Unfortunately, we did not collect data on the previous colonization of our MRSA BSI cases. Rapid microbiological diagnosis could also be useful for the identification of MRSA in blood cultures as it identifies the presence of resistance genes but unfortunately does not provide information on their expression and MIC of antibiotics. Therefore, it may be particularly useful in the presence of local epidemiological data (such as those collected in our center) to decide on initial (empirical) MRSA antibacterial therapy, which should be modified, if necessary, when MICs of different anti-MRSA antibiotics are available. The main limitations of the study are due to the retrospective and anonymous data collection and relate specifically to the unavailability of data on the presence of deep organ localization, the inaccurate definition of risk factors for BSI, and the fact that the final outcome was analyzed very roughly, only in terms of patients’ death. However, we believe that the final favorable outcome may be related to the use of drugs other than vancomycin, when necessary. In fact, the analysis of hospital causes of death, during the study period, reported no deaths due to MRSA bacteremia.

In conclusion, this study allowed us to become aware of two important events: the increase in BSI episodes due to MRSA with an increased MIC for vancomycin and the high proportion of MRSA BSI in which vancomycin AUC_0–24_/MIC ratio does not achieve a therapeutic value (≥400) despite dose changes. These observations could be important for antimicrobial stewardship programs and for medical training of young clinicians regarding knowledge, attitudes, and practices on the use of antibiotics in the management of antibiotic-resistant pathogens, including MRSA [22]. Medical training plays a central role in combating antimicrobial resistance that is one of the major health issues worldwide.

Further studies are needed to investigate the presumed relationship between observed changes in the consumption of specific antibiotics and changes in the epidemiology of specific pathogens and infections.

## Figures and Tables

**Figure 1 antibiotics-12-00216-f001:**
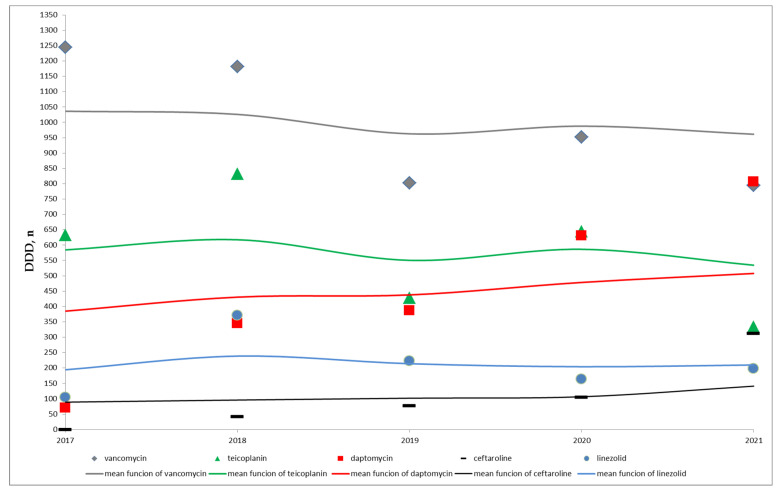
Time distribution of the number of antibiotic DDDs and of the respective mean functions- non parametric local-constant kernel regressions.

**Figure 2 antibiotics-12-00216-f002:**
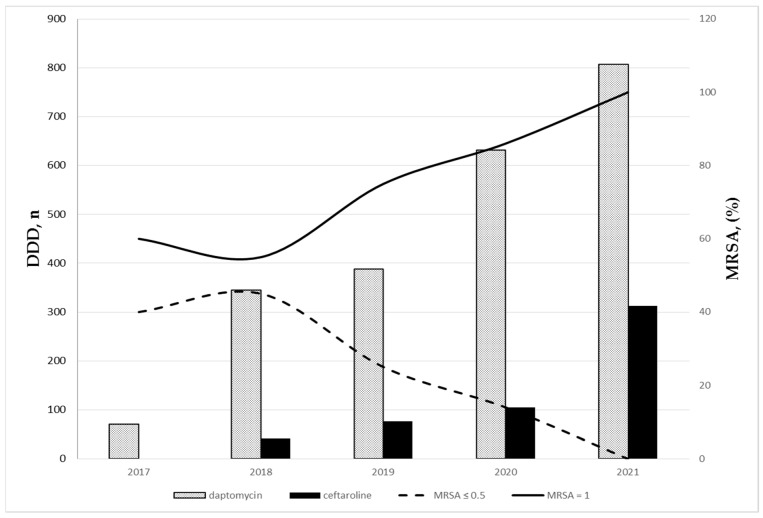
Time distribution of the number of daptomycin and ceftaroline DDDs and of the percentage of MRSA BSIs according to vancomycin MIC.

**Table 1 antibiotics-12-00216-t001:** Distribution of the antibiotic consumption and of the MRSA BSIs over the 5-year period.

Year	2017	2018	2019	2020	2021	Total	*p*-Value	Kendall’s Rank Correlation Coefficient
	Total amount of delivered antibiotics, g DDD, n					Median (IQR)		
Vancomycin, DDD: 2 g	2488.5 1244	2363.5 1182	1603 802	1903.5 952	1587.5 794	1903.5 (1603–2363.5) 952 (802–1182)	0.086 ^1^ 0.086 ^2^	−0.800
Teicoplanin, DDD: 0.4 g	253 633	332.8 832	171.4 429	257.8 645	133.4 334	253 (171.4–257.8) 633 (429–645)	0.340 ^1^ 0.426 ^2^	−0.400
Daptomycin, DDD: 0.28 g	19.6 70	96.2 344	108.4 387	176.6 631	225.9 807	108.4 (96.2–176.6) 387 (344–387)	0.057 ^1^ 0.027 ^2^	1.000
Ceftaroline, DDD: 1.2 g	0	46.6 41	92.4 77	126 105	373.8 312	92.4 (46.6–126) 77 (41–105)	0.057 ^1^ 0.027 ^2^	1.000
Linezolid, DDD: 1.2 g	126 105	444.6 371	267.6 223	196.8 164	237 198	237 (196.8–267.6) 198 (164–223)	0.849 ^1^ 1.000 ^2^	0.000
	MRSA BSI, n (%)							
Vancomycin MIC ≤ 0.5 mg/L	2 (40.0)	5 (45.4)	2 (25.0)	2 (14.3)	0	11 (24.4)	0.017 ^3^	-
Vancomycin MIC = 1 mg/L	3 (60.0)	6 (54.5)	6 (75.0)	12 (85.7)	7 (100)	34 (75.6)		-
Total	5 (100)	11 (100)	8 (100)	14 (100)	7 (100)	45 (100)		

DDD = defined daily dose; MRSA = methicillin resistant *S. aureus*; MIC= minimally inhibitor concentration. ^1^ Nonparametric test for trend across ordered groups developed by Cuzik; ^2^ nonparametric Kendall’s rank correlation test; ^3^ chi-square statistic for the trend of proportions.

## Data Availability

Data are contained within the article. They are available on request from the corresponding author.

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
