# Peer review of "Changes in the Use of Antibiotics for Methicillin-Resistant Staphylococcus aureus Bloodstream Infections in Children: A 5-Year Retrospective, Single Center Study"

_antibiotics, 2023, doi:10.3390/antibiotics12020216_

Round 1
Reviewer 1 Report
This is an interesting study including a long period of five years. I appreciate the tables, figures and references. Therefore, I believe this study adds to the body of literature in the field of MRSA. Only have two suggestions for the authors:
1) add limitation section
2) improve English language
Author Response
Response to Reviewer 1 Comments - manuscript No.: 2141823
Point 1: … add limitation section.
Response 1: We thank the Reviewer for the positive feedback. A limitation section has been added in the discussion before the paragraph of the conclusions (line 143 in the tracked changes file).
Point 2: improve English language
Response 2:
We have carefully reviewed the manuscript to improve English language
Reviewer 2 Report
Authors wrote an very interesting paper, well presenting and with good idea research. Below my minor suggestions:
1. Introduction: add that Italy has the worste incidence of MDR infection in Europe and also for this issue focus on MRSA is crucial to improve Antimicrobial resistance spread. In addiction, add how SARS-CoV2 pandemic has caused an direct and indirect increase of AMR with disruption of healthcare systems and practices (see and cite Impact of SARS-CoV-2 Epidemic on Antimicrobial Resistance: A Literature Review. Viruses. 2021 Oct 20;13(11):2110. doi: 10.3390/v13112110.)
2. Methods: well done
3. Results: clear, I appreciate it
4. Discussion: discuss also the use of fast microbiology as a strategy to control MRSA and recuse inappropriate prescription and also the possible role of nasopharyngeal swabs to assess respiratory colonisation by MRSA.
In addiction, add the role of education of youg doctor in AMR to improve the management and the prescription of antibiotics (see and cite Italian young doctors' knowledge, attitudes and practices on antibiotic use and resistance: A national cross-sectional survey. J Glob Antimicrob Resist. 2020 Dec;23:167-173. doi: 10.1016/j.jgar.2020.08.022.)
Please give some global health action that came from your very interesting and well wrote paper.
Author Response
Response to Reviewer 2 Comments - manuscript No.: 2141823
Point 1: … Introduction: add that Italy has the worst incidence of MDR infection in Europe and also for this issue focus on MRSA is crucial to improve Antimicrobial resistance spread. In addiction, add how SARS-CoV2 pandemic has caused an direct and indirect increase of AMR with disruption of healthcare systems and practices (see and cite Impact of SARS-CoV-2 Epidemic on Antimicrobial Resistance: A Literature Review. Viruses. 2021 Oct 20;13(11):2110. doi: 10.3390/v13112110.)
Response 1: We thank the Reviewer for this important comment, the following sentence “This may be of utmost importance in countries such as Italy where the incidence of antibiotic-resistant infections is high, and has worsened during COVID-19 pandemics (Segala FV, Bavaro DF, Di Gennaro F, Salvati F, Marotta C, Saracino A, Murri R, Fantoni M. Impact of SARS-CoV-2 Epidemic on Antimicrobial Resistance: A Literature Review. Viruses. 2021 Oct 20;13(11):2110. doi: 10.3390/v13112110), although this aspect has not been observed in Italian children (Garazzino S, Lo Vecchio A, Pierantoni L, Calò Carducci FI, Marchetti F, Meini A, Castagnola E, Vergine G, Donà D, Bosis S, Dodi I, Venturini E, Felici E, Giacchero R, Denina M, Pierri L, Nicolini G, Montagnani C, Krzysztofiak A, Bianchini S, Marabotto C, Tovo PA, Pruccoli G, Lanari M, Villani A, Castelli Gattinara G; Italian SITIP-SIP Pediatric Infection Study Group. Epidemiology, Clinical Features and Prognostic Factors of Pediatric SARS-CoV-2 Infection: Results From an Italian Multicenter Study. Front Pediatr. 2021 Mar 16;9:649358. doi: 10.3389/fped.2021.649358.)” has been added in the Introduction section. This comment was also reported in the discussion (line 226 in the tracked changes file). The references, accordingly, have been updated with the new two references.
Point 2: Discussion: discuss also the use of fast microbiology as a strategy to control MRSA and recuse inappropriate prescription and also the possible role of nasopharyngeal swabs to assess respiratory colonisation by MRSA.
Response 2:
In the discussion, before the section on limitations (line 232 in the tracked changes file) we have added the comment that rapid microbiological diagnosis can be important for early diagnosis of colonized patients and can also be useful for identification of MRSA in blood cultures. It identifies the presence of resistance genes but does not provide information on their expression and MIC of antibiotics. Therefore, it may be particularly useful in the presence of local epidemiological data to decide on initial (empirical) MRSA antibacterial therapy, which should be modified, if necessary, when MICs of different anti-MRSA antibiotics are available.
Point 3: In addiction, add the role of education of young doctor in AMR to improve the management and the prescription of antibiotics (see and cite Italian young doctors' knowledge, attitudes and practices on antibiotic use and resistance: A national cross-sectional survey. J Glob Antimicrob Resist. 2020 Dec;23:167-173. doi: 10.1016/j.jgar.2020.08.022.)
Response 3:
We have added in the conclusions (line 253 in the tracked changes file) a sentence about the role of medical training of young clinicians and the references, accordingly, have been updated with the new reference (Di Gennaro F, Marotta C, Amicone M, Bavaro DF, Bernaudo F, Frisicale EM, Kurotschka PK, Mazzari A, Veronese N, Murri R, Fantoni M. Italian young doctors' knowledge, attitudes and practices on antibiotic use and resistance: A national cross-sectional survey. J Glob Antimicrob Resist. 2020 Dec;23:167-173. doi: 10.1016/j.jgar.2020.08.022).
Point 4: Please give some global health action that come from your very interesting and well wrote paper.
Response 4:
Accordingly to the previous point, we have added in the conclusions the comment regarding the role of medical training of young clinicians which plays a central role to combat antimicrobial resistance, one of the major health issues worldwide.
Reviewer 3 Report
The authors documented the increasing Vancomycin resistance in MRSA bloodstream isolates (BSI) from children, and the increasing utilization of alternative antibiotics replacing Vancomycin.
The BSI in children are serious conditions, most of the cases develop as a consequence of a malignant disease (leukemia) or other immunocompromised conditions (immaturity) The number of children included in the study are 45 without indicating their age and the underlying disease leading the development of BSI. The number of patients (age, sex, and underlying condition) could be included in the “Materials and Methods”.
Author Response
Response to Reviewer 3 Comments - manuscript No.: 2141823
Point 1: … The BSI in children are serious conditions, most of the cases develop as a consequence of a malignant disease (leukemia) or other immunocompromised conditions (immaturity) The number of children included in the study are 45 without indicating their age and the underlying disease leading the development of BSI. The number of patients (age, sex, and underlying condition) could be included in the “Materials and Methods”.
Response 1:
We thank the Reviewer for this important observation. We agree that some details regarding patients with MRSA BSI were to be reported. We have now supplemented the results (lines 146-151 and 166 in the tracked changes file) with available data on age and ward of admission as surrogate of underlying disease and accordingly we have also modified the section on Materials and Methods.